# Lower Plasma Melatonin Levels Predict Worse Long-Term Survival in Pulmonary Arterial Hypertension

**DOI:** 10.3390/jcm9051248

**Published:** 2020-04-25

**Authors:** Zongye Cai, Theo Klein, Laurie W. Geenen, Ly Tu, Siyu Tian, Annemien E. van den Bosch, Yolanda B. de Rijke, Irwin K. M. Reiss, Eric Boersma, Dirk J. Duncker, Karin A. Boomars, Christophe Guignabert, Daphne Merkus

**Affiliations:** 1Department of Cardiology, Erasmus MC, University Medical Center Rotterdam, 3000 CA Rotterdam, The Netherlands; z.cai@erasmusmc.nl (Z.C.); l.geenen@erasmusmc.nl (L.W.G.); s.tian@erasmusmc.nl (S.T.); a.e.vandenbosch@erasmusmc.nl (A.E.v.d.B.); h.boersma@erasmusmc.nl (E.B.); d.duncker@erasmusmc.nl (D.J.D.); 2Department of Clinical Chemistry, Erasmus MC, University Medical Center Rotterdam, 3000 CB Rotterdam, The Netherlands; t.klein@erasmusmc.nl (T.K.); y.derijke@erasmusmc.nl (Y.B.d.R.); 3INSERM UMR_S 999, Hôpital Marie Lannelongue, Le Plessis-Robinson, 92350 Paris, France; lyieng@gmail.com (L.T.); guignabert@gmail.com (C.G.); 4Université Paris-Saclay, School of Medicine, Le Kremlin-Bicêtre, 94270 Paris, France; 5Department of Pediatrics/Neonatology, Sophia Children’s Hospital, Erasmus MC, University Medical Center Rotterdam, 3000 CB Rotterdam, The Netherlands; i.reiss@erasmusmc.nl; 6Department of Clinical Epidemiology, Erasmus MC, University Medical Center Rotterdam, 3000 CA Rotterdam, The Netherlands; 7Department of Pulmonary Medicine, Erasmus MC, University Medical Center Rotterdam, 3000 CA Rotterdam, The Netherlands; k.boomars@erasmusmc.nl; 8Walter Brendel Center of Experimental Medicine (WBex), LMU Munich, 81377 Munich, Germany; 9German Center for Cardiovascular Research (DZHK), Partner Site Munich, Munich Heart Alliance (MHA), 81377 Munich, Germany

**Keywords:** melatonin, pulmonary hypertension, survival, clinical outcome

## Abstract

Exogenous melatonin has been reported to be beneficial in the treatment of pulmonary hypertension (PH) in animal models. Multiple mechanisms are involved, with melatonin exerting anti-oxidant and anti-inflammatory effects, as well as inducing vasodilation and cardio-protection. However, endogenous levels of melatonin in treatment-naïve patients with PH and their clinical significance are still unknown. Plasma levels of endogenous melatonin were measured by liquid chromatography-tandem mass spectrometry in PH patients (*n* = 64, 43 pulmonary arterial hypertension (PAH) and 21 chronic thromboembolic PH (CTEPH)) and healthy controls (*n* = 111). Melatonin levels were higher in PH, PAH, and CTEPH patients when compared with controls (Median 118.7 (IQR 108.2–139.9), 118.9 (109.3–147.7), 118.3 (106.8–130.1) versus 108.0 (102.3–115.2) pM, respectively, *p* all <0.001). The mortality was 26% (11/43) in the PAH subgroup during a long-term follow-up of 42 (IQR: 32–58) months. Kaplan–Meier analysis showed that, in the PAH subgroup, patients with melatonin levels in the 1st quartile (<109.3 pM) had a worse survival than those in quartile 2–4 (Mean survival times were 46 (95% CI: 30–65) versus 68 (58–77) months, Log-rank, *p* = 0.026) with an increased hazard ratio of 3.5 (95% CI: 1.1–11.6, *p* = 0.038). Endogenous melatonin was increased in treatment-naïve patients with PH, and lower levels of melatonin were associated with worse long-term survival in patient with PAH.

## 1. Introduction

Pulmonary hypertension (PH) is a severe disease with a wide spectrum of underlying etiologies [1]. Pulmonary arterial hypertension (PAH) and chronic thromboembolic PH (CTEPH) are two subgroups of PH, that show severe pulmonary vascular remodeling. Current treatment strategies delay disease progression, but curative treatment, reversing microvascular remodeling, has not been established [2,3]. Therefore, research identifying novel mechanisms of disease progression and identifying potential therapeutic targets is necessary to develop new therapeutic strategies and improve prognosis.

Melatonin is a hormone mainly synthesized by the pineal gland and is well-known for its role in the regulation of circadian rhythm [4]. Over the past decades, an increasing number of studies demonstrated that exogenous melatonin also exerts protective effects in cardiovascular diseases [5,6,7], respiratory diseases [8], and cancers [9]. It was already shown in 2007 that chronic hypoxia induced PH was associated with the loss of the pulmonary vasorelaxation effect of melatonin [10], while supplementation of melatonin could prevent chronic hypoxia induced PH via anti-proliferative and anti-inflammatory effects [11,12], as well as through inhibiting oxidative stress [13,14,15], restoring nitric oxide production [11], and increasing angiogenesis [16]. These beneficial effects of melatonin were also shown in the rat models of monocrotaline-induced PH [12,17,18], and Sugen-hypoxia-induced PH [12]. In addition, melatonin was also found to be cardio-protective in monocrotaline-induced PH by improving RV function and inhibiting cardiac fibrosis [16].

Although animal studies suggest that exogenous melatonin might be beneficial for patients with PH, endogenous melatonin levels in animal models of PH, and in treatment-naïve patients with PH and their clinical significance are still unknown. In the present study, we therefore tested the hypothesis that lower melatonin levels would be associated with poor prognosis in PH patients. For this purpose, we investigated plasma melatonin levels in two well-established rat models of PH, and in treatment-naïve patients with PH and studied their clinical significance.

## 2. Methods

### 2.1. Study Population

A total of 64 consecutive treatment-naïve adult patients with PH, including 43 patients with PAH (Group 1) and 21 patients with CTEPH (Group 4), diagnosed by right heart catheterization according to the guidelines between May 2012 and October 2016 were included as PH group in this prospective observational cohort study [19,20]. Exclusion criteria for PH group were: incomplete diagnostic procedure, not treatment-naïve, not capable of signing informed consent, and other Groups of PH, including some patients from Group 1 PH, and all patients from Group 2, 3, and 5 PH (Figure 1). A healthy control group consisting of 145 self-reported healthy volunteers, without any (prior) cardiovascular diseases and risk factors, was recruited during the same period via an advertisement for healthy subjects, 34 volunteers were excluded from this study because of a blood pressure over 140/90 mmHg at the time of visit. More details about the study design of both cohorts have been previously described [21,22]. The study protocols were approved by the Erasmus MC Ethical Committee and written informed consent was obtained by all PH patients and healthy volunteers. All procedures were performed in accordance with Declaration of Helsinki.

### 2.2. Follow-Up of PH Patients

PH patients were prospectively followed-up until 1 January 2019. All patients were prescribed with specific PAH medications and/or treated with balloon pulmonary angioplasty or pulmonary endarterectomy (CTEPH patients) when indicated according to the guidelines [19,23]. The primary endpoint was defined as all-cause mortality. Survival status of all patients was obtained from patients and checked in the Municipal Personal Records database. Patients who did not reach the primary endpoint were censored at the 1 January 2019.

### 2.3. Animal Models of PH

Two well-established rat models of severe PH were used in this study as previously described [24]. In brief, a monocrotaline-induced PH model (*n* = 11) was established in 4 weeks-old male Wistar rats (Janvier Labs, Saint Berthevin, France) with a single subcutaneous injection of monocrotaline (40 mg/kg, Sigma-Aldrich, Saint-Quentin-Fallavier, France) for 3 weeks. The Sugen-hypoxia-induced PH model (*n* = 10) was established in 4 weeks-old Wistar rats (Janvier Labs, Saint Berthevin, France) with a single subcutaneous injection of Sugen (SU5416, 20 mg/kg, Sigma-Aldrich, Saint-Quentin-Fallavier, France) combined with exposure to normobaric hypoxia for 3 weeks followed by room air for 5 weeks.

### 2.4. Blood Sampling and Measurement of Melatonin

For PH patients, regular peripheral venous blood sampling was performed during the diagnostic right heart catheterization. For healthy volunteers, regular peripheral venous blood sampling was performed at the time of visit. For animal models of PH, blood sampling was performed before sacrifice. All blood sampling was conducted during daytime between 9:00 and 18:00, in which period the levels of melatonin were reported to be stable [25]. All blood samples were prepared as EDTA-plasma samples, and then frozen and stored in aliquots at −80 °C, and thawed only once for use. Melatonin levels were measured using ultra-performance liquid chromatography-tandem mass spectrometry (UPLC-MS/MS). Briefly, 10 µL plasma was mixed with 10 µL deuterated melatonin (Melatonine-D3, Buchem BV, Apeldoorn, The Netherlands) solution as internal isotopically labeled internal standard and subsequently mixed with 80 µL acetonitrile for protein precipitation. After 10 min, the samples were cleared by centrifugation and 90 µL supernatant was dried under a stream of nitrogen at 30 °C. The residue was reconstituted in 40 µL H_2_O (in-house purified using a Milli-Q device) with 0.2% (*v*/*v*) formic acid before quantitative analysis for melatonin using an in-house developed UPLC-MS/MS assay on a Sciex 6500+ QTRAP mass spectrometer (Sciex, Nieuwerkerk ad Ijssel, The Netherlands) hyphenated to a Shimadzu Nexera UPLC system (Shimadzu Benelux, Den Bosch, The Netherlands). Ten microliters of reconstituted sample was resolved on an Acquity HSS T3 UPLC column (2.1 × 100 mm, 1.8 µm; Waters, Etten-Leur, The Netherlands) using a gradient of acetonitrile in Milli-Q water, each with 0.2% formic acid (*v*/*v*). Melatonin was detected in MRM mode by the mass transition m/z 233.1/174.1 (DP 56 V, CE 20V) and quantified to a standard calibration curve of 50– pM using the area ratio of melatonin/melatonin-D3.

## 3. Statistical Analysis

Data were tested for adherence to a normal distribution with the Kolmogorov–Smirnov method. Continuous variables are presented as mean ± standard deviation (SD) or median (interquartile range (IQR)), categoric variables as numbers (percentages), or as otherwise reported. Group comparisons of continuous variables (e.g., melatonin levels, age) were performed using the unpaired *t*-test or Mann–Whitney test (2 groups, e.g., human PH versus controls, Rat models of PH versus controls), and one-way ANOVA or Kruskal–Wallis Test (3 groups, human controls, PAH, and CTEPH). Groups comparisons of categoric variables (e.g., sex, NYHA) were performed using the chi-square test. Correlations analysis between melatonin levels and baseline characteristics were determined using the Spearman correlation coefficient. Logistic regression was conducted to determine whether plasma melatonin was an independent risk factor that distinguishes between PH patients and healthy controls. Univariate and multivariate Cox proportional hazard regression were used to assess associations between plasma levels of melatonin and mortality in PAH patients, one PAH patient with a very high melatonin level of 4471 pM was defined as an outlier (>100 times the IQR), and was excluded to avoid interference. Comparisons of long-term survival curves between groups in PAH patients were performed using Kaplan–Meier analysis with log-rank (for trend) test. Statistical analysis was performed using IBM SPSS software (version 21.0.0.1), figures were made using GraphPad Prism (version 8.0.2). A two-sided *p*-value < 0.05 was considered statistically significant.

## 4. Results

### 4.1. Baseline Characteristics

Baseline characteristics of all treatment-naïve patients with PH, PAH, CTEPH, and healthy controls are summarized in Table 1. PH patients were older and had higher heart rate and body mass index than controls. PAH patients showed more severe PH than CTEPH patients.

### 4.2. Levels of Plasma Melatonin

The median of plasma melatonin levels in healthy volunteers was 108.0 (102.3–115.2) pM, and was higher in treatment-naïve patients with PH (118.7 (108.2–139.9) pM, *p* < 0.001), PAH (118.9 (109.3–147.7) pM, *p* < 0.001) and CTEPH (118.3 (106.8–130.1) pM, *p* < 0.01) (Figure 2A). There was no difference between patients with PAH and CTEPH, and there was no sex difference in either controls or PH patients. In addition, melatonin levels were significantly higher in rat models of monocrotaline-induced PH (148.0 (107.2–175.8) pM, *p* < 0.01) and Sugen-hypoxia-induced PH (103.2 (83.7–118.1) pM, *p* < 0.01) as compared to the control rats (67.6 (58.9–80.2) pM), and were similar in these two rat models of PH (Figure 2B).

### 4.3. Correlation Analysis

In healthy controls, there was a weak association between melatonin and heart rate (r = −0.229, *p* = 0.016), while no association was seen between melatonin with age, sex, body mass index, and systolic blood pressure (Table 2).

In patients with PH, melatonin was inversely associated with age (r = −0.368, *p* = 0.003) and systolic blood pressure (r = −0.251, *p* = 0.046). In the subgroup of patients with PAH, melatonin was also inversely associated with age (r = −0.334, *p* = 0.029). No association was seen in patients with CTEPH (Table 2).

Neither in patients with PH, nor in the subgroups of PAH and CTEPH, a correlation was found between melatonin levels with hemodynamic parameters (mean pulmonary artery pressure, pulmonary artery wedge pressure, pulmonary vascular resistance) and cardiac function (cardiac output, cardiac index, the 6-min walk distance, NYHA class) (Table 2).

### 4.4. Logistic Regression Analyses

Logistic regression analysis was performed to determine whether plasma melatonin was an independent risk factor that distinguishes PH patients and controls. Before correction for the potential confounders (age, sex, and body mass index), plasma melatonin distinguished PH patients and controls (Odds Ratio 1.035 (95% CI 1.016–1.055), *p* < 0.001), PAH patients and controls (1.036 (1.016–1.057), *p* < 0.001), CTEPH patients and controls (1.029 (1.002–1.056), *p* = 0.033) (Table 3). However, after correction for potential confounders, although plasma melatonin still distinguished PH patients and controls, it only distinguished PAH patients but not CTEPH patients and controls (Table 3). These results indicated that plasma melatonin was only an independent risk factor for PAH, but not for CTEPH.

### 4.5. Long-Term Survival Analyses

During a median follow-up time of 42 (32–58) months, 12 patients (11 PAH patients and 1 CTEPH patient) reached the primary endpoint, the observed mortality rates were 19% (12/64) in the total PH group, 26% (11/43) in the PAH subgroup, and 5% (1/21) in the CTEPH subgroup. Long-term survival analysis was performed in the PAH subgroup.

Initially, PAH patients were stratified into 4 groups according to the quartiles of melatonin levels in the PAH subgroup: 1st quartile < 109.3 pM, 2nd quartile from 109.3 to 118.9 pM, 3rd quartile from 118.9 to 147.7 pM, 4th quartile > 147.7 pM. The mortality in these 4 groups was 55% (6/11), 10% (1/10), 0% (0/12), and 40% (4/10), respectively (Figure 3). No significant difference in the survival curves was observed among the 4 groups (Log-rank for trend, *p* = 0.478, Figure 4A). However, patients in the 1st quartile and 4th quartile seemed to have worse survival than others. Therefore, when considering melatonin levels as continuous variable in Cox proportional hazard analysis, there was no significant association between melatonin levels and mortality, without or with adjustment for age, sex, and body mass index (Table 4).

We next undertook a two-group survival comparison based on quartiles of melatonin levels. Kaplan–Meier analyses showed that there was no significant difference between patients with melatonin levels below and above the median (118.9 pM, Log-rank, *p* = 0.449, Figure 4B). Similarly, stratifying patients based on melatonin levels within and below the 4th quartile showed no difference in survival (147.7 pM, Log-rank, *p* = 0.122, Figure 4C). However, patients with melatonin levels in the 1st quartile (<109.3 pM) had a worse long-term cumulative survival than patients with melatonin levels in the 2nd to 4th quartile (mean survival times were 46 (95% CI: 30–65) versus 68 (95% CI: 58–77) months, Log-rank, *p* = 0.026, Figure 4D) with a significant increased hazard ratio of 3.529 (95% CI: 1.070–11.642, *p* = 0.038).

When looking at baseline characteristics, patients with melatonin levels in the 1st quartile were older than others, while there was no difference in other characteristics (Table 5). After adjustment for age in the Cox model, the hazard ration of death for low melatonin levels was no longer significant (1.607 (95% CI: 0.402–6.426), *p* = 0.503)), suggesting that age may be a potential confounding variable.

## 5. Discussion

The present study demonstrates, for the first time, that plasma melatonin levels at the time of diagnosis predict clinical outcome in patients with PAH. The main findings of the study are that plasma melatonin levels were higher in treatment-naïve patients with PH than in healthy controls, which was supported by the findings in two experimental models of PH. Higher levels of melatonin were an independent risk factor of PAH in logistic regression analysis. However, lower levels of melatonin were predictive of worse long-term survival for PAH patients.

Our study demonstrates that plasma melatonin levels were higher in treatment-naïve patients with PH when compared with healthy individuals, as well as in two rat models with monocrotaline- or Sugen-hypoxia-induced PH as compared to control rats, suggesting an increase in melatonin in PH. These data seem to be in contrast with a recent study, showing that melatonin was decreased in serum from PAH patients [12]. An important difference with our study is that the cohort in the study of Zhang et al. was small (15 PAH patients versus 8 controls), and that the patients were treated with PAH medication whereas our patients were treatment-naïve.

The pathophysiological mechanisms underlying higher levels of melatonin in treatment-naïve patients with PH versus healthy controls are unclear. It has been shown that melatonin levels decline with age in healthy humans [26]. In contrast, such a negative correlation was absent in the healthy controls in our cohort, but was present in PH patients. As PH patients, that were generally older, had even higher levels of melatonin than the younger controls, the difference between PH and controls is unlikely to be caused by the age difference.

Melatonin is mainly produced by the pineal gland [4], and is an important regulator of circadian rhythm [27,28]. Therefore, an entire 24 h profile of melatonin levels, with knowledge sleeping patterns, is preferable to describe the melatonin levels, with samples taken under a strict light control (<10 lux) because of the strong direct suppressive effect of light on melatonin synthesis in the pineal gland. However, melatonin production in the pineal gland, which increases at night, is stable during daytime and shows seasonal variation [25,29]. Since we did not observe a seasonal sampling effect on melatonin levels in either healthy controls or PH patients (data not shown) and higher melatonin levels were also present in two rat models of PH, which were housed in the same facility with identical light–dark cycles, we believe that the increased melatonin represents a feature of disease.

In addition to synthesis in the pineal gland, the enzymes that convert serotonin into melatonin, serotonin N-acetyltransferase and N-Acetylserotonin O-methyltransferase, were found to be present not only in the pineal gland but also in the plasma [30] and the lung [31]. Melatonin synthesis can be activated by the activation of sympathetic system and renin-angiotensin system [32,33,34,35]. PH patients and the two rat models of PH have previously been shown to exhibit increased sympathetic activity (consistent with a higher heart rate in PH patients in the present study) as well as activation of the renin-angiotensin system [36,37,38,39,40]. Moreover, serotonin, the precursor of melatonin, is increased in the plasma of patients with pulmonary hypertension [41]. These may have contributed to the higher levels of melatonin in plasma.

Although there are differences in the pathophysiology between PAH and CTEPH, endothelial dysfunction and pulmonary vascular remodeling are common features for both subgroups [2]. Interestingly, melatonin levels were increased in both PAH and CTEPH patients, but did not correlate with hemodynamic parameters or cardiac functional severity, as evidenced by a lack of correlation with pulmonary artery pressure, pulmonary vascular resistance, cardiac output, 6MWD, and NYHA class. Although melatonin levels were similarly elevated in patients with PAH and CTEPH, melatonin appeared only as an independent risk factor for PAH but not for CETPH in logistic regression. In addition, 11 out of 43 patients with PAH died, whereas all CTEPH patients except one survived. This might be attributed to the fact that PAH patients showed a more severe PH phenotype with higher pulmonary vascular resistance than CTEPH patients, and indicates that melatonin is not simply a reflection of disease severity. Importantly, in PAH patients, lower melatonin levels were associated with a worse long-term survival although age may be a confounding factor in this association. Worse survival with lower melatonin levels is in accordance with a recent study showing that lower levels of melatonin in patients with dilated cardiomyopathy correlated with a poor prognosis, worse cardiac function (lower cardiac output) and more cardiac injury (i.e., higher levels of troponin T) [6]. Furthermore, several studies show cardiovascular protective effects of exogenous melatonin in both humans and animal models [5,6,7,42,43], suggesting that higher endogenous melatonin levels may exert a protective effect in PH. Indeed, melatonin induces vasodilation, has anti-proliferative effects as well as antioxidant and anti-inflammatory properties [11,12,13,14,15,16,17,18,44], thereby counteracting the vasoconstriction, excessive cell proliferation, increased oxidative stress, and inflammatory infiltration characteristic of PH [3]. We therefore propose that PAH patients with endogenous melatonin in the lowest quartile may have lost the benefits of its protective effects. Importantly, a protective effect of exogenous melatonin is still present in the rat models of PH [12,17,18] as utilized in the present study, despite the fact that our study shows that the endogenous levels of melatonin were already increased in these models. Conversely, PAH patients with the highest levels of endogenous melatonin seemed to have a high mortality in the present study. We believe that these high endogenous melatonin levels may be attributed to hyper-activation of sympathetic system and/or renin-angiotensin system [32,33,34,35], which are present in severe PAH patients and therefore contribute to a poor survival [40,45,46]. Therefore, whether exogenous melatonin supplements may be effective as a therapeutic strategy in patients with PH remains to be established.

## 6. Conclusions

To our knowledge, this is the first prospective cohort study demonstrating that lower levels of plasma melatonin at the time of diagnosis predict worse long-term survival in PAH patients, however, whether exogenous melatonin supplements may be effective as a therapeutic strategy in human PH remains to be established.

## Figures and Tables

**Figure 1 jcm-09-01248-f001:**
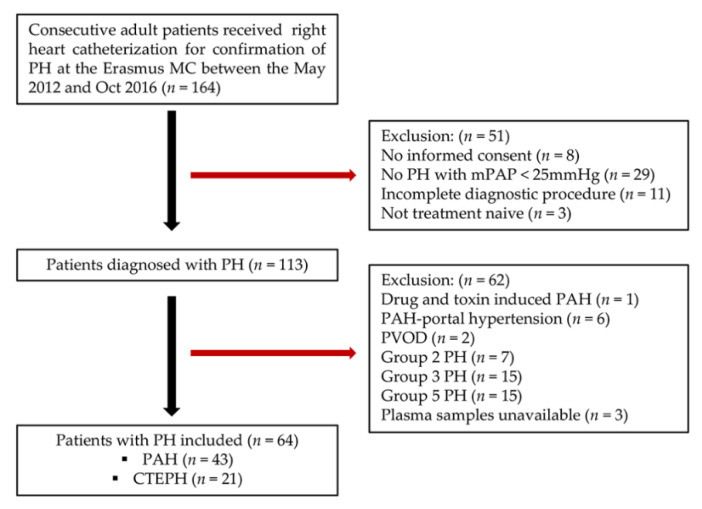
Enrollment scheme of PH patients in the current study. PH: pulmonary hypertension, mPAP: mean pulmonary artery pressure, PAH: pulmonary arterial hypertension, PVOD: pulmonary veno-occlusive disease, CTEPH: chronic thromboembolic pulmonary hypertension.

**Figure 2 jcm-09-01248-f002:**
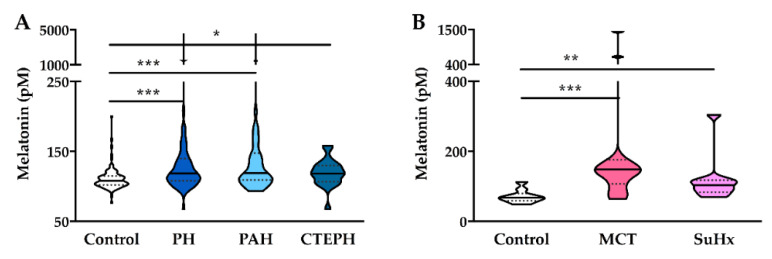
Plasma melatonin was increased in patients with PH and 2 rat models of PH. (**A**) Plasma melatonin was higher in patients with PH (*n* = 64), PAH (*n* = 43), and CTEPH (*n* = 21) than in healthy controls (*n* = 111), but there was no difference between PAH and CTEPH. (**B**) Plasma melatonin was higher in 2 rat models of PH, including MCT-induced PH (*n* = 11) and SuHx-induced PH (*n* = 10), than in controls (*n* = 9), but there was no difference between these two models. Distribution of the Data was shown in violin plots with median (solid line) and interquartile range (dotted lines). * *p* < 0.05, ** *p* < 0.01, *** *p* < 0.001, Mann–Whitney Test or Kruskal–Wallis Test. PH: pulmonary hypertension; PAH: pulmonary arterial hypertension; CTEPH: chronic thromboembolic PH; MCT: monocrotaline; SuHx: sugen and hypoxia.

**Figure 3 jcm-09-01248-f003:**
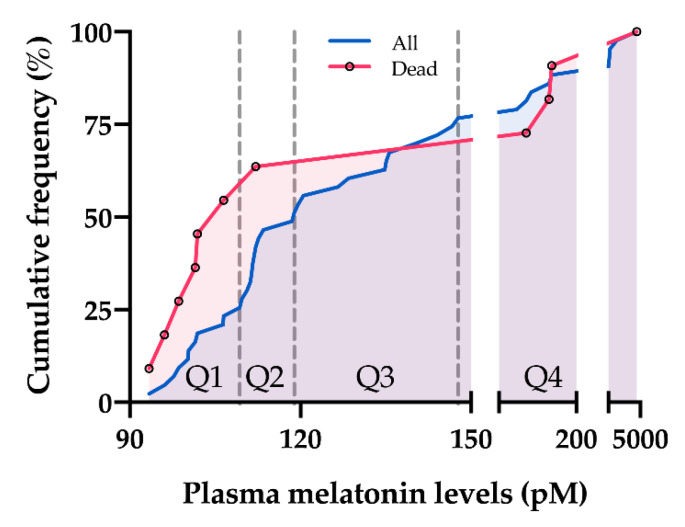
Distribution of mortality in PAH patients. PAH patients were stratified into 4 groups according to the quartiles of melatonin levels in PAH patients: 1st quartile (Q1) < 109.3 pM, 2nd quartile (Q2) from 109.3 to 118.9 pM, 3rd quartile (Q3) from 118.9 to 147.7 pM, 4th quartile (Q4) > 147.7 pM. The mortality per quartile was 55% (6/11), 10% (1/10), 0% (0/12), and 40% (4/10), respectively.

**Figure 4 jcm-09-01248-f004:**
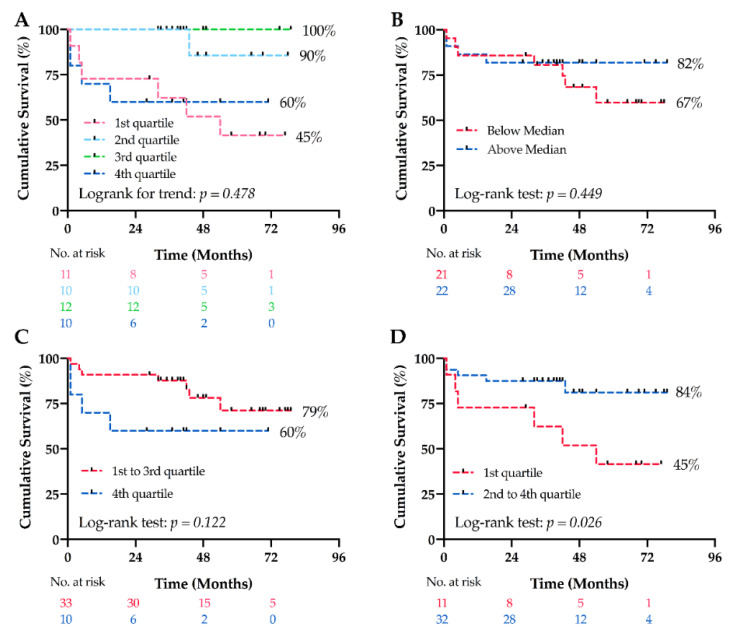
Long-term survival analysis in PAH patients. (**A**) There were no significant differences in long-term survival among 4 quartiles stratified according to melatonin levels in PAH patients. (**B**) There was no significant difference in long-term survival between patients with melatonin levels below and above the median (118.9 pM). (**C**) There was no significant difference in long-term survival between patients with melatonin levels in the 4th quartile (>147.7 pM) as compared to quartile 1–3. (**D**) Patients with melatonin levels in the 1st quartile (<109.3 pM) had a worse long-term cumulative survival than patients with melatonin levels in quartile 2–4.

**Table 1 jcm-09-01248-t001:** Baseline characteristics.

	Control	PH		
		Total	PAH	CTEPH
*n*	111	64	43	21
Aetiology				
iPAH, *n* (%)			15 (35)	
CTD-PAH, *n* (%)			17 (40)	
CHD-PAH, *n* (%)			11 (25)	
Age, years old	43 ± 13	55 ± 17 ***	53 ± 17 **	58 ± 18 ***
Sex, women *n* (%)	59 (53)	41 (64)	29 (67)	12 (57)
sBP, mmHg	123 (115–128)	127 (115–136)	122 (114–132)	133 (124–141) **^,†^
HR, beats·min ^−1^	68 (62–76)	78 (65–90) **	78 (67–90) **	71 (61–88)
BMI, kg·m ^−2^	23.8 ± 2.9	28.4 ± 6.3 ***	27.0 ± 6.1 ***	31.4 ± 5.7 ***
mPAP, mmHg	-	46.8 ± 15.7	50.5 ± 16.1	39.3 ± 12.3 ^††^
PAWP, mmHg	-	12.4 ± 5.1	11.8 ± 5.6	13.7 ± 3.3
PVR, WU	-	5.8 (3.3–9.8)	7.1 (5.1–11.8)	3.4 (3.0–5.3) ^††^
CO, L·min ^−1^	-	5.0 (4.1–5.9)	4.7 (3.9–5.5)	5.4 (4.7–6.4) ^†^
CI, L·min ^−1^·m ^−2^	-	2.6 (2.3–3.2)	2.5 (2.2–3.3)	2.7 (2.3–3.0)
6MWD, m	-	353 ± 146	337 ± 153	385 ± 130
NYHA, 1:2:3:4	-	1:25:31:7	1:13:23:6	0:12:8:1

Data was present as mean ± SD, median (IQR), or numbers (percentages). ** *p* < 0.01, *** *p* < 0.001 versus control; †*p* < 0.05, ††*p* < 0.01 versus PAH. Student T Test, Mann–Whitney U Test, one-way ANOVA, Kruskal–Wallis Test, or chi-square Test. PH: pulmonary hypertension; PAH: pulmonary arterial hypertension; CTEPH: chronic thromboembolic PH; iPAH: idiopathic PAH; CTD-PAH: connective tissues diseases associated PAH; CHD-PAH: congenital heart diseases associated PAH; sBP: systolic blood pressure; HR: heart rate; BMI: body mass index; mPAP: mean pulmonary arterial pressure; PAWP: pulmonary arterial wedge pressure; PVR: pulmonary vascular resistance; CO: cardiac output; CI: cardiac index; 6MWD: 6-min walking distance; NYHA: New York Heart Association classification.

**Table 2 jcm-09-01248-t002:** Correlations between plasma levels of melatonin and baseline characteristics.

	Plasma Levels of Melatonin							
	Control		PH		PAH		CTEPH	
	r	*p* Value	r	*p* Value	r	*p* Value	r	*p* Value
Baseline characteristics								
Age	−0.119	0.212	**−0.368**	**0.003**	**−0.334**	**0.029**	−0.363	0.106
Sex	−0.070	0.466	0.103	0.417	0.112	0.475	0.159	0.491
sBP	−0.178	0.063	**−0.251**	**0.046**	−0.279	0.070	−0.015	0.949
HR	**−0.229**	**0.016**	0.088	0.488	0.155	0.321	−0.182	0.430
BMI	−0.025	0.796	−0.162	0.201	−0.140	0.372	−0.018	0.938
mPAP			0.166	0.191	0.061	0.699	0.403	0.070
PAWP			−0.028	0.841	−0.039	0.820	0.178	0.509
PVR			0.094	0.518	0.097	0.584	0.091	0.737
CO			−0.184	0.160	−0.154	0.351	−0.302	0.184
CI			−0.185	0.158	−0.170	0.301	−0.339	0.133
6MWD			0.103	0.459	0.164	0.340	−0.057	0.823
NYHA			0.029	0.821	0.033	0.832	−0.084	0.717

Significant correlations are shown in bold. PH: pulmonary hypertension; PAH: pulmonary arterial hypertension; CTEPH: chronic thromboembolic PH; sBP: systolic blood pressure; HR: heart rate; BMI: body mass index; mPAP: mean pulmonary arterial pressure; PAWP: pulmonary arterial wedge pressure; PVR: pulmonary vascular resistance; CO: cardiac output; CI: cardiac index; 6MWD: 6-min walking distance; NYHA: New York Heart Association classification.

**Table 3 jcm-09-01248-t003:** Logistic regression analyses of plasma melatonin to distinguish PH patients and controls.

		Univariate	Multivariate ^#^	
			Model 1	Model 2
PH	Odds Ratio (95% CI)	1.035 (1.016–1.055)	1.048 (1.022–1.074)	1.047 (1.021–1.073)
	*p* value	<0.001	<0.001	<0.001
PAH	Odds Ratio (95% CI)	1.036 (1.016–1.057)	1.049 (1.022–1.076)	1.047 (1.020–1.074)
	*p* value	<0.001	<0.001	<0.001
CTEPH	Odds Ratio (95% CI)	1.029 (1.002–1.056)	1.025 (0.989–1.062)	1.025 (0.988–1.062)
	*p* value	0.033	0.175	0.184

^#^ Model 1 was adjusted for age, and body mass index. Model 2 was adjusted for age, sex, and body mass index. CI: confidential interval.

**Table 4 jcm-09-01248-t004:** Cox proportional hazard analysis for death per pM increase in melatonin in PAH patients.

Analyses	Hazard Ratio (95% CI)	*p* Value
Univariate	0.995 (0.981–1.010)	0.546
Multivariate ^#^		
Model 1	0.999 (0.992–1.005)	0.653
Model 2	0.998 (0.992–1.005)	0.645

^#^ Model 1 was adjusted for age, and body mass index. Model 2 was adjusted for age, sex, and body mass index. CI: confidential interval.

**Table 5 jcm-09-01248-t005:** Baseline characteristics in PAH patients in and above the 1st quartile of melatonin levels.

	PAH		
	1st Quartile (<109.3 pM)	Quartile 2–4 (≥109.3 pM)	*p* Value
*n*	11	32	
Aetiology			
iPAH, *n* (%)	2 (18)	13 (41)	
CTD-PAH, *n* (%)	6 (55)	11 (34)	
CHD-PAH, *n* (%)	3 (27)	8 (25)	
Age, years old	66 ± 13	48 ± 15	0.001
Sex, women *n* (%)	9 (82)	20 (63)	0.213
sBP, mmHg	127 ± 14	123 ± 15	0.466
HR, beats·min ^−1^	79 ± 14	80 ± 18	0.965
BMI, kg·m ^−2^	27.1 ± 3.9	26.9 ± 6.8	0.931
mPAP, mmHg	47.0 (38.0–65.0)	45.0 (38.8–65.3)	0.880
PAWP, mmHg	13.0 ± 5.1	11.3 ± 5.8	0.450
PVR, WU	5.7 (3.9–11.4)	8.8 (5.6–11.9)	0.316
CO, L·min ^−1^	5.1 ± 1.5	4.8 ± 1.4	0.522
CI, L·min ^−1^·m ^−2^	2.9 ± 0.8	2.6 ± 0.7	0.312
6MWD, m	271 ± 148	356 ± 152	0.172
NYHA, 1:2:3:4	0:4:4:3	1:9:19:3	0.359

Data are presented as mean ± SD, median (IQR), or numbers (percentages). Student T Test, Mann–Whitney U Test, or chi-square test were used for comparison. PAH: pulmonary arterial hypertension; iPAH: idiopathic PAH; CTD-PAH: connective tissues diseases associated PAH; CHD-PAH: congenital heart diseases associated PAH; sBP: systolic blood pressure; HR: heart rate; BMI: body mass index; mPAP: mean pulmonary arterial pressure; PAWP: pulmonary arterial wedge pressure; PVR: pulmonary vascular resistance; CO: cardiac output; CI: cardiac index; 6MWD: 6-min walking distance; NYHA: New York Heart Association classification.

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
