# Peer review of "Lower Plasma Melatonin Levels Predict Worse Long-Term Survival in Pulmonary Arterial Hypertension"

_jcm, 2020, doi:10.3390/jcm9051248_

Round 1

Reviewer 1 Report

The authors report data from a prospective cohort correlating melatonin levels with long-term survival on pulmonary hypertensive patients (n=64) between 2012 and 2019. The patients were divided according to their PH classification as PAH (n=43) and CTEPH (n=21). 145 health subjects were recruited as control. Clinical characteristics and melatonin levels were collected at baseline between 2012 and 2016. In addition, the authors also measured melatonin levels in two animal models of PH. Higher melatonin plasma levels were found among PH groups versus controls in both clinical and experimental studies. After adjustment for potential confounders, melatonin levels were higher only in PAH versus controls. Survival analysis showed that PAH patients with melatonin levels below 109.3pM (1st quartile) had worse long-term survival.

The manuscript has interesting findings and it was carefully designed. A few comments/ questions arise.

Minor comment: The authors exposed the limitation that no causality can be inferred regarding the use of exogenous melatonin supplements in PH. However, as they have available two animal models of PH, interventional studies in these animals would strength the current manuscript.

Question: What is the severity of PH and age of patients in the 1st quartile  (<109.3pM) compared to the other groups (>109.3pM)? An additional table comparing clinical characteristics such as age, sex, BMI and severity of PH of 1st quartile group vs other quartiles should be included. This is crucial to support the current conclusion.

Author Response

The manuscript has interesting findings and it was carefully designed. A few comments/ questions arise.

Minor comment: The authors exposed the limitation that no causality can be inferred regarding the use of exogenous melatonin supplements in PH. However, as they have available two animal models of PH, interventional studies in these animals would strength the current manuscript.

Reply: We would like to thank the reviewer for his/her positive comments. Although our groups has not performed the studies as requested, interventional studies in these , as well as other animal models of PH have performed. We have elaborated a bit more on these studies in the introduction (line66-68) as well as the discussion (Line 369-376).  

Question: What is the severity of PH and age of patients in the 1st quartile (<109.3pM) compared to the other groups (>109.3pM)? An additional table comparing clinical characteristics such as age, sex, BMI and severity of PH of 1st quartile group vs other quartiles should be included. This is crucial to support the current conclusion.

Reply: As outlined in the original manuscript, melatonin levels indeed decrease with age in patients with PH. We have added table 5 (line 299-308) describing the baseline characteristics of patients with melatonin levels below and above the 1stquartile.This Table shows that patients with melatonin levels below the 1stquartile were older than others, while there was no difference in other baseline characteristics (line 293-297). These data are consistent with the finding that melatonin is declining with aging in PAH patients in the present study, and previous study in healthy controls and show that age may be a potential confounding variable as has now been mentioned in the revised manuscript (line 363).

Reviewer 2 Report

This is an interesting manuscript that provides some preliminary correlations with plasma melatonin levels and clinical patient outcomes. 

It should be noted that plasma melatonin levels follow a circadian pattern and controlling for the time of visit may refine these results and provide more compelling evidence for these effects. 

In addition, although interesting - there is no clear mechanism that can be responsible for these effects. Therefore, presenting the potential confounding variables (such as adequate sleep, daytime versus evening/night work) might also strengthen the conclusions made.

Overall, an interesting manuscript which outlines a previously unknown finding. 

Author Response

Comment: This is an interesting manuscript that provides some preliminary correlations with plasma melatonin levels and clinical patient outcomes.

It should be noted that plasma melatonin levels follow a circadian pattern and controlling for the time of visit may refine these results and provide more compelling evidence for these effects.

Reply: We would like to thank the reviewer for his/her positive comments. Unfortunately, we have no data on the exact time of blood sampling. However, it has been shown that within the circadian pattern of plasma melatonin levels, which peaks at night, melatonin levels are relatively stable during  daytime between 9:00 and 18:00, in which period the blood sampling was performed (methodology line 121-122, reference 25 of the revised manuscript). We have acknowledged this limitation in the discussion (line 333-337).

Comment: In addition, although interesting - there is no clear mechanism that can be responsible for these effects. Therefore, presenting the potential confounding variables (such as adequate sleep, daytime versus evening/night work) might also strengthen the conclusions made.

Reply: Unfortunately, we have no data on sleep and/or night time labor in our patient cohort.  However, we believe that the elevated melatonin levels may not be a consequence of synthesis of melatonin in the pineal gland, but rather in the lungs of PH patients and therefore represents a consequence of the disease. This is supported by the observation that melatonin was also elevated in our animal model, in which the animals with PH were housed under identical conditions as compared to control animals (discussion, line 338-341).

Comment: Overall, an interesting manuscript which outlines a previously unknown finding.

Reply: Thank you!